# Acute Pancreatitis—Drivers of Hospitalisation Cost—A Seven-Year Retrospective Study from a Large Tertiary Center

**DOI:** 10.3390/healthcare11182482

**Published:** 2023-09-07

**Authors:** Mihai Radu Pahomeanu, Dalia Ioana Constantinescu, Irina Ștefania Diaconu, Dana Gabriela Corbu, Lucian Negreanu

**Affiliations:** 1Faculty of Medicine, Carol Davila University of Medicine and Pharmacy, 050474 Bucharest, Romania; 2Department of Gastroenterology and Internal Medicine, University Emergency Hospital of Bucharest, 050098 Bucharest, Romania

**Keywords:** acute pancreatitis, costs, severity, daily cost, financial burden

## Abstract

(1) Introduction: Acute pancreatitis (AP) remains a global burden of cost for healthcare services. We found a high degree of heterogeneity in cost-related reports and a scarcity of data regarding the cost of AP episodes in European and Asian populations. We aimed to estimate the median daily cost of hospitalisation (DCH) of AP in our population. Our secondary aims included estimating the total cost of hospitalisation (TCH) and the total cost of AP in Romania, as well as assessing the correlation between median DCH and ward, age, sex, length of stay (LoS), intensive care unit (ICU), outcome, severity, morphology, and aetiology of AP. (2) Material and methods: This retrospective cohort study included 1473 cases recruited from the electronic health records of the University Emergency Hospital of Bucharest. Statistical tests used included Kolmogorov–Smirnov, Kruskal–Wallis with post-hoc Dunn–Bonferroni, and Pearson correlation two-tailed. (3) Results: We found a median DCH of AP of USD 203.8 and a median TCH of USD 1360.5. The total yearly cost of AP in Romania was estimated at around USD 19 million. The majority of males with AP (61.8%) were mostly discharged as healed/ameliorated (83.8%); a majority had local complications (55.4%), which were mostly alcohol-related (35.1%). Regarding the aetiology, biliary-related AP was a cost driver, with significant statistical differences observed in all studied groups (*p* < 0.01). Morphology assessment revealed that acute necrotic collections were associated with high cost and meaningful disparities among the groups (*p* < 0.01). Cost was also associated with severity, with significant deviations among all groups (*p* < 0.01). Outcome-at-discharge as deceased correlated with higher costs, with substantial differences within groups (*p* < 0.01). The need for an intensive care unit was also a large driver of cost (*p* < 0.01). Females were prone to more expensive costs (*p* < 0.01). Surgical cases necessitated more financial resources (*p* < 0.01). (4) Conclusions: To the best of our knowledge, this is the first study on the cost of AP in Romania. Our findings showed that the drivers of increased AP costs might be older age, ICU, intra-hospital mortality, severe AP, local complications such as acute necrotic collections, biliary aetiology, and female sex. We found large heterogeneity and scarcity regarding cost-related data in the literature.

## 1. Introduction

### 1.1. Context

Acute pancreatitis (AP) represents a major burden of cost for all healthcare services worldwide. The Global Burden of Disease Study 2019 estimated 2.8 million cases of AP worldwide annually, with a little over 14,000 cases/year and an estimated incidence of 50.8 cases/100,000 persons per year in Romania [1]. Regarding the analysis of cost in AP, there is a high degree of interstate heterogeneity in the U.S.A., and a scarcity of European and Asian data in the literature. We did not find any previous studies regarding the costs of AP episodes in Romania.

In 2007, Fagenholtz et al. [2], a study that included 344,000 cases, reported a total annual cost of hospitalised AP in the U.S.A. at USD 2.2 billion, with a mean total charge of hospitalisation (TCH) of USD 24,000, and a mean daily cost of hospitalisation (DCH) of USD 1670. A high heterogeneity of costs was reported within the U.S.A., with the highest mean TCH of USD 18,795.34 in Vermont and the lowest mean TCH of USD 4873.99 in Kansas. Another U.S. study with a population of more than 280,000 cases by Yeh et al. [3] observed a TCH of USD 12,446.48 in 2014. In 2022, Peery et al. [4] showed that AP was the second most common gastroenterological diagnosis in U.S. hospitals, only after upper gastrointestinal bleeding, with a rising trend in the two decades, with a median TCH of USD 22,817. In 2013, Andersson et al. [5] reported an annual cost of handling AP in Sweden of USD 51 million and a mean TCH of USD 26,222. A 2018 meta-analysis from Spain led by F. Valverde-Lopez [6] found a mean DCH of USD 175.3, whereas the Murata et al. [7] 2012 retrospective study based on the Japanese population reported a national TCH of USD 67.2 million, a mean TCH of USD 9343.4, and a mean DCH of USD 484.5.

These findings underscore the high discrepancies in the field of cost analysis in hospitalised AP, which may be related to the different types of health systems examined. However, there is a scarcity of data from Eastern Europe, especially from Romania, thus stimulating calls for action from fellow researchers, such as Li et al. [1], regarding the country.

### 1.2. Aims and Objectives

In a medium-income region such as Eastern Europe, poor financial resources allocation might hamper the evidence-based medicine management of patients, as often enforced budget caps might prevent patients from receiving their required healthcare services. Hence, it is highly important, in our opinion, that policymakers have access to clear data based on the real-life national/regional situation. Given that AP is a frequently encountered pathology in both gastroenterological and surgical departments, both worldwide and in Eastern Europe, it represents a large driver of costs in these wards. Notably, few studies have been performed in Eastern Europe to debate this topic. Thus, we sought to offer a cornerstone to support healthcare decision makers in the informed and evidence-based allocation of material resources based on a case-by-case assessment. Given regional variances in laws, genetics, cultural habits, and, most importantly, the epidemiology of AP, we consider this study of high interest for our region/country.

In this study, we aimed to estimate the median DCH of AP in a large tertiary centre in southern Romania. The secondary aims of our study were to estimate the median TCH and total cost of AP cases in Romania and test the possible correlation between median DCH and independent factors, such as ward of care (gastroenterological or surgical), sex, intensive care unit (ICU) admission, outcome, severity, morphology, aetiology, length of stay (LoS), and age.

## 2. Materials and Methods

### 2.1. Case Selection

Cases were selected from the electronic health records (EHRs) of the University Emergency Hospital of Bucharest (Romanian: Spitalul Universitar de Urgență București) as consecutive cases of AP admitted to our hospital from 1 April 2017 to 1 April 2022. The University Emergency Hospital of Bucharest, founded in 1978, is the largest acute-care teaching tertiary hospital in Romania, accounting for 1099 beds, and has both gastroenterological and abdominal surgery departments.

This retrospective study was approved by the Institutional Review Board (IRB) of our hospital. General informed consent for using their medical data in retrospective and non-interventional studies was obtained on behalf of the patients prior to their hospitalisation. This study follows the ethical guidelines of the 1975 Declaration of Helsinki and the STROBE guidelines. All patients were 18 years old or older at the time of hospitalisation.

We queried the EHRs through ICD-10 codes (K85, B26.3 and B25.2), obtaining 1525 consecutive hospitalisations. After the sample was screened by a team of medically trained staff, we observed that 52 (3.4%) were miscoded, so we eliminated them. For this study, any new hospitalisation of the same patient was considered a new case. To our knowledge, this is the largest population of AP published to date in Romania.

Due to the scarcity of cases in our study, we chose to exclude cases with walled-off necrosis (WON) (*n* = 4) and those with stationary discharge outcomes (*n* = 7) from the comparison of the groups. We found no case of aggravated outcomes at discharge, thus, we did not take them into account.

### 2.2. Currency Exchange and Cost-Related Values Methodology

Costs were extracted from discharge documents available in the hospital’s EHRs, as calculated by the hospital’s accounting department. These costs represent the sum of the following factors: costs of hospitalisation (LoS multiplied by daily hospitalisation fixed rate of our hospital), food allowance (LoS multiplied by daily food allowance), drug costs, medical materials used cost, paraclinical investigation costs and other investigations. It is important to state that the following are regulated through law (Romanian Government Resolution nr. 696/2021) by the national public insurer and Ministry of Health (according to several indexes particular to every tertiary centre, such as number of beds and case complexity index for the previous year): daily hospitalisation fixed rate, daily food allowance, paraclinical investigation cost, and other investigation costs. Given that most of the studies regarding the costs of AP originated from the U.S.A. and had costs reported in USD, we reported our cost-related values in the same currency.

From the EHRs, we obtained values of cost in Romanian Leu (RON) that were converted to USD values at the exchange rate of RON to USD on 1 April 2022. Due to a high heterogeneity regarding the date of discharge, we were not able to adjust for inflation or to temporally use adequate exchange rates. For references [5,6,8,9] that used EUR to report cost-related values, we used the exchange rate of EUR to USD from the date they were received by the journal: 15 August 2013, 7 April 2018, 8 March 2008, and 17 March 2014, respectively. The DCH was calculated as TCH divided by the LoS in days.

For the historical exchange rate, we used the online software provided by Oanda FX Data Services ^®^ “https://www.oanda.com/currency-converter (accessed on 31 May 2023)”. All cost-related values in the Appendix A are the original ones expressed in RON.

### 2.3. Diagnosis of AP

We considered a case of AP to be any hospitalisation that fulfilled at least two out of three criteria needed to diagnose AP (three times normal values of amylasemia and/or lypasemia, specific abdominal pain and/or imagistic criteria).

### 2.4. Methodology for Aetiology Stratification

Aetiology was stratified as stated in *Sleisenger and Fordtran’s gastrointestinal and liver disease*, 10th edition [10]. For ease of reading and using statistical tests, we addressed only the four most frequent aetiologies found in the EHRs (Table 1), classifying the rest for which we had data as ‘others’. In cases of mixed aetiology involving at least one of the first four, we reported the cases for only one of those aetiologies based on author consensus; otherwise, they were classified as ‘other’.

We used the IAP/APA guidelines [11] criteria in considering any of the most often encountered aetiologies (alcoholic, biliary), as the rest were classified on author consensus: alcoholic—as described by the case physician through anamnesis (assessing alcohol intake); biliary—alanine aminotransferase (ALT) > 150 U/L within 48 h after onset of symptoms, together with ultrasonographic signs of gallstones (dilation of common bile ducts > 5 mm, gallstones, and/or sludge present in the 1/3 distal common bile duct, and/or known gallstone disease); hypertriglyceridemia—serum triglycerides >750 mg/dL (in at least one instance while hospitalised); decompensated diabetes mellitus—fasting serum glycemia > 250 mg/dL (in at least two different measurements), HbA1C > 7.5% (in at least one instance), and no other more probable cause.

### 2.5. Methodology for Defining Groups

Details regarding the cases’ characteristics that were included in this study are summarised in Table 1. Severity and morphology were stratified as in the Revised Atlanta Classification [12]. In all cases, morphology was assessed by abdominal ultrasound or CT scan if patients failed to respond to conservative treatment, there was any doubt regarding the diagnosis and aetiology, or the medical team sought a confirmation of clinical assessment of severity based on IAP/APA guidelines [11]. Outcome was based on EHRs, as they were reported by the case physician. In comparing the groups, we also excluded those with idiopathic aetiology, those classified as having other aetiology, and those from which we did not have any data regarding morphology or rurality.

Cases were attributed to gastroenterological or surgical wards based on the decision of a multidisciplinary medical board (that included the following on-call specialists: emergency room physician, gastroenterologist, general surgeon and/or radiologist) based on the evidence-based medicine guidelines of IAP/APA [11] and clinical judgement in particular situations not explicitly mentioned in the aforementioned guide.

### 2.6. Statistical Analysis

The dataset was organised using Microsoft Office Excel 2019 © (now known as Microsoft 365 ©) and Google Docs ©. For analysis of the general characteristics of the cohort, found in Table 1, we used frequency tests. To check for homogeneity of distribution, we used the Kolmogorov–Smirnov test. In the case of non-normal distribution of population for comparison of medians, we used the Kruskal–Wallis test with the post hoc Dunn–Bonferroni test to check for differences within groups when there were more than two. The Pearson correlation test (two-tailed) was used to check the correlation between the two continuous variables. All statistical tests were run on IBM SPSS Statistics version 29.0.0.0 ©. For statistical significance, we reported *p* = < 0.05. We reported the third decimal on the *p*-value only in cases where the value was close to those aforementioned.

All the values from the figures were reported in the original currency (RON—Romanian Leu).

## 3. Results

### 3.1. Population Characteristics

In our population, we found a majority of male (*n* = 911, 61.8%) cases, mostly cared for in a gastroenterological ward (*n* = 860, 58.4%), mostly of mild evolution (*n* = 758, 51.8%) and discharged as healed or ameliorated (*n* = 1234, 83.8%). A total of 816 (55.4%) were found to have local complications, mostly alcoholic (*n* = 517, 35.1%) and biliary (*n* = 509, 34.6%) aetiologies. A few cases were admitted to the ICU (*n* = 125, 8.5%) (see details in Table 1 and Appendix A.

### 3.2. General Costs

Median DHC was reported at USD 203.8 (IQR = 95.5) and median TCH at USD 1360.5 (IQR = 1241.6). Based on the Global Burden of Disease Study 2019, as reported by Li et al. [1], estimates of 14,037.2 cases annually of AP in Romania, and our estimates regarding median TCH, we estimated a total burden of AP in Romania at more than USD 19 million (USD 19,097,610.6) annually.

### 3.3. Aetiology

We used the Kolmogorov–Smirnov test to assess the homogeneity of distribution of the four groups and found that none had a normally distributed population pertaining to the median DCH. We obtained the following values: for alcoholic aetiology, D(517) = 0.17 and *p* < 0.01; for biliary, D(509) = 0.48 and *p* < 0.01; for hypertriglyceridemia, D(80) = 0.26 and *p* < 0.01; and for diabetes mellitus, D(62) = 0.32 and *p* < 0.01 (see Appendix A for results and graphs).

The Kruskal–Wallis test conducted to compare the median DCH showed significant differences between all groups, with the exception of hypertriglyceridemia and diabetes mellitus (H(3) = 239.91, *p* < 0.01). Using post-hoc Dunn–Bonferroni tests, we observed statistically significant differences between alcoholic and biliary (Z = −324.59, *p* < 0.01, adj. *p* < 0.01), alcoholic and hypertriglyceridemic (Z = −119.38, *p* < 0.01, adj. *p* < 0.01), biliary and diabetes mellitus (Z = 216.45, *p* < 0.01, adj. *p* < 0.01), and between biliary and hypertriglyceridemic (Z = 205.22, *p* < 0.01, adj. *p* < 0.01) aetiologies. There was an insignificant difference between alcoholic and diabetes mellitus aetiologies (Z = −108.14, *p* = 0.02, adj. *p* = 0.10) (Figure 1, for median DCH stratified by groups; see Table 1 and Appendix A for detailed results).

### 3.4. Morphology

The Kolmogorov–Smirnov test used to assess heterogeneity of distribution revealed a normally distributed population in the interstitial (D(60) = 0.10, *p* = 0.20) and acute necrosis collection (ANC) (D(63) = 0.11, *p* = 0.08) groups. Regarding the other groups considered for comparison, interstitial (D(575) = 0.16, *p* < 0.01), APFC (D(114) = 0.21, *p* < 0.01) and normal pancreas (D(210) = 0.50, *p* < 0.01) had a heterogeneity of distribution of cases regarding median DCH (see Appendix A for results and graphs). To compare the median DCH within groups, we used the Kruskal–Wallis test and found statistically significant differences (H(4) = 24.35, *p* < 0.01) between interstitial and APFC (Z = −87.19, *p* < 0.01, adj. *p* = 0.04), interstitial and ANC (Z = −138.43, *p* < 0.01, adj. *p* < 0.01), interstitial and normal pancreas (Z = −69.22, *p* < 0.01, adj. *p* = 0.036), and pseudocyst and acute necrotic collection (Z = −161.41, *p* < 0.01, adj. *p* = 0.02) groups. No other significant differences were found between the groups. For median DCH stratified by groups, see Figure 2 and Table 1. See Appendix A for detailed results and graphs.

### 3.5. Severity

The tests used for assessing heterogeneity of distribution revealed that none of the groups were normally distributed in relation to median DCH, given the following results: mild (D(758) = 0.48, *p* < 0.01), moderately severe (D(542) = 0.46, *p* < 0.01), and severe (D(173) = 0.23, *p* < 0.01) groups. Statistically significant differences between groups were found using Kruskal–Wallis tests (H(2) = 93.68, *p* < 0.01). Dunn–Bonferroni post-hoc test revealed statistically significant differences between all groups, specifically between mild and moderately severe (Z = −96.70, *p* < 0.01, adj. *p* < 0.01), mild and severe (Z = −343.07, *p* < 0.01, adj. *p* < 0.01), and moderately severe and severe (Z = −246.37, *p* < 0.01, adj. *p* < 0.01) disease groups. For median DCH stratified by group, see Figure 3 and Table 1. Appendix A presents detailed results and graphs.

### 3.6. Outcome at Discharge

We did not find any of the groups we considered to be normally distributed with respect to median DCH. Specifically, the Kolmogorov–Smirnov test revealed healed/ameliorated (D(1234) = 0.47, *p* < 0.01); transferred (H(54) = 0.52, *p* < 0.01), discharge-at-will (D(101) = 0.14, *p* < 0.01), and deceased (D(77) = 0.14, *p* < 0.01) groups. We found statistical differences regarding median DCH between all groups (H(3) = 126.05, *p* < 0.01), with the exception of the comparison between healed/ameliorated and discharge-at-will (Z = −100.88, *p* = 0.13) groups. We found the following values in post hoc testing: healed/ameliorated and transfer (Z = −273.22, *p* < 0.01, adj. *p* < 0.01); healed/ameliorated and deceased (Z = −514.38, *p* < 0.01, adj. *p* < 0.01); transfer and deceased (Z = −241.16, *p* < 0.01, adj. *p* < 0.01); and discharge-at-will and deceased (Z = −413.45, *p* < 0.01, adj. *p* < 0.01) groups. There was no statistical difference found in the comparison of the discharge-at-will and transfer groups (Z = 172.33, *p* = 0.02, adj. *p* = 0.094). For median DCH stratified by groups, see Figure 4 and Table 1. Appendix A presents detailed results and graphs.

### 3.7. ICU

With regard to ICU admission, we found that both groups had a heterogeneous distribution of cases. Specific values revealed by the Kolmogorov–Smirnov test were D(1348) = 0.47, *p* < 0.01 for those that did not necessitate ICU, and D(125) = 0.24, *p* < 0.01 for those that did. For these two aforementioned groups, the Kruskal–Wallis test revealed statistically significant differences with regard to median DCH (H(1) = 187.59, *p* < 0.01), with ICU cases showing a higher median DCH (USD 309.0 vs. USD 194.3). See Table 1 and Figure 5. Appendix A presents detailed results and graphs.

### 3.8. Sex

Concerning the median DCH, both the male (D(911) = 0.19, *p* < 0.01) and female (D(562) = 0.47, *p* < 0.01) groups were not normally distributed. There was a significant statistical difference between the two groups (H(1) = 54.53, *p* < 0.01), with the female group showing a higher median DCH (USD 223.0 vs. 188.7) (Figure 6 and Table 1). See Appendix A for detailed results and graphs.

### 3.9. Gastroenterological versus Surgical Cases

Regarding the ward in which the cases were treated and median DHC, we found that both gastroenterological (D(860) = 0.22, *p* < 0.01) and abdominal surgery (D(613) = 0.47, *p* < 0.01) wards were not normally distributed. Comparing median DCH between the two groups, we found a statistically significant difference (H(1) = 728.61, *p* < 0.01), with surgical cases showing a higher DCH (USD 256.5 vs. 167.3), as shown in Figure 7.

### 3.10. Length of Stay (LoS) and Age

In our study population, we found a median age of 56 years (IQR = 25) and a median LoS of 7 days (IQR = 5). Using the Pearson correlation test, we found a statistically significant but very low positive correlation between age and DCH (*r* = 0.06, *p* = 0.02, *n* = 1473) and no statistically significant correlation between LoS and DCH (*r* = −0.03, *p* = 0.26, *n* = 1473). See Appendix A for details.

## 4. Discussion

Our study of cost in AP involved a multiple-layer stratification of the independent variables regarding ward of care, sex, ICU, outcome at discharge, severity, and morphology (both as described by Atlanta Revised Classification [12]) and aetiology. To our knowledge, this is one of the few studies regarding the cost of AP in Europe conducted in the last decade and the first endeavour of this kind in Romania.

We found a median DCH in a tertiary centre from southern Romania of USD 203.8 and a median TCH of USD 1360.5, with an estimation of the total cost of AP in Romania at around USD 19 million yearly. Our values were 16 times lower than the one reported in the U.S.A. in 2022 by Peery et al. [4] regarding TCH, although the large degree of interstatal heterogeneity of cost as compared by Yeh et al. [3] in 2014 put Kansas at less than four times higher than our TCH. Comparing our median DCH with the one reported by Fagenholtz et al. [2] in the U.S.A. in 2007, again the differences were substantial, at eight times higher. Regarding the comparison with the Japanese results from a study conducted in 2012 [7], our estimates of DCH were only two-fold higher. This high heterogeneity of results, at least compared with ours, was probably a result of the difference in healthcare insurance systems and the level of income per capita between Romania on the one hand, and, on the other hand, the U.S.A. and Japan, two high-income countries. This difference could indicate the availability of larger financial resources for the latter, implying better access to healthcare for American and Japanese citizens.

We also weighed our estimates against those of our fellow European researchers with whom we share a high stake of the public sector in the healthcare industry, highly centralised public-owned healthcare insurers, and a high degree of cultural and genetical similarity regarding our populations. We found lower DCH in Spain (USD 175.3) as reported by a study from 2018 [6], but an almost seven times higher TCH in Sweden, as reported in 2013 by Andersson et al. [5]. Although differences persist even within our continent, those seem to be negligible in comparison with Romania, as a country with a higher GDP per capita than ours, namely Spain, manages to have a lower DCH than us. These differences might reside in a more thoughtful financial allocation of the healthcare systems in Spain, or they might simply be a result of the different methodologies used by our colleagues in determining DCH. The high heterogeneity in cost-related studies methodology regarding AP and the diversity of structures regarding the healthcare insurer and the organisation of healthcare systems around the world, coupled with a scarcity of studies in this field, still makes it difficult to reasonably compare costs across countries. In this regard, we consider that there exists a necessity for a methodology guideline for cost studies, at least in the pancreatology field.

In this study, we observed statistically significant higher costs for surgical cases (USD 256.5 vs. USD 167.3) in comparison with those with gastroenterological disease. The lower DCH of gastroenterological cases might originate from the lower cost of endoscopic procedures in comparison with classical surgical ones, as shown in a randomised clinical trial published in 2018 [13] that compared the mean TCH of cases treated for infected necrotising pancreatitis with minimally invasive surgery techniques, compared with the endoscopic step-up approach. This hypothesis should imply that, at least financially, there is a benefit in transitioning from a surgical resolution to an endoscopic resolution, where there is a possibility. With current advances not only in technology but also in the geographical distribution of gastroenterological centres that can provide cutting-edge procedures, such as endoscopic retrograde cholangiopancreatography or endoscopic ultrasound, this transition is easier to achieve. Regarding the discrepancy between severity stratification and ICU admission, as there was a lower rate of admission in ICU than all severe cases, it is noteworthy that our study captured the 2020–2021 COVID-19 pandemic. This unforeseeable and saddening event heightened addressability to ICU departments far above the capacity, and some severe cases needed to be addressed in normal care wards.

Regarding the cost of ICU, we found substantial differences in median DCH between cases not admitted to ICU and those who were (USD 194.3 vs. USD 309.0). Further, we found higher median TCH in those who went through the ICU (USD 1278.2 vs. USD 3937.9). We found a fivefold lower TCH in our study compared with a study published in 2023 [14] on the cost of AP in ICU settings of multiple centres from Australia and New Zealand. Compared with a Finnish study [8] on ICU resources spending in AP published in 2008, we found a very high discrepancy as a 33-times lower cost (USD 3937.9 vs. USD 133,587) and an almost tenfold lower cost than median DCH reported in the U.K. in 2014 in a small retrospective study by Durrani et al. (USD 309.0 vs. USD 3066.5) [9]. Concerning the ICU cost, the probable limit of our study resides in the fact that we formed an estimation of the ICU cost based on hospital expenditures without being able to take into account separate financing from the national intensive care programme.

Nevertheless, as the ICU seems to be a large driver of costs worldwide, there should be better profiling of the cases that require ICU admission, as there should be an improvement in the management of AP right from the first hours of hospitalisation. Several attempts to develop a better way to fight the most important mechanism through which organ failure is provoked, namely systemic inflammatory response syndrome, were sought, none of them offering a clear-cut result.

On the subject of cost stratification relating to the discharge outcome, we found the lowest median DCH for those healed or ameliorated (USD 195.8), and the highest in the deceased group (USD 469.4). There were statistically substantial disparities between the healed/ameliorated group and the deceased and transferred cases (Md DCH = USD 243.7). The deceased case group also had statistically meaningful variations regarding transfer and discharge-at-will (Md DCH = USD 211.1). We did not find data that assessed AP cost by stratification regarding discharge outcomes. However, these results were in congruence with what we had expected, as the great drivers for mortality in AP—such as systemic inflammatory response syndrome, shock [15,16], multiple organ failure (respiratory, cardiovascular, or renal) [17], and infections [18]—require more investigations, more expensive molecules and/or more procedures to treat. We believe that the high cost of the deceased cases was driven not only by frailty but also a higher rate of comorbidity of those cases, as one of the limitations of our study was not accounting for those, as well as by the higher efforts undertaken by their case physician to limit the evolution of the disease.

Concerning severity stratification, we found the lowest median DCH in the mild disease group (USD 191.2) and the highest in the severe (SAP) group (USD 256.7). There were statistically significant variances between the three groups. These data were similar to the findings by Murata et al. [7], which in a study published in 2012 showed that the most expensive cases were those with SAP courses, with four times greater DCH compared with our study. Further, Andersson et al. [5] found the highest cost in SAP (TCH 12 times higher costs compared with ours), and Pokras et al. [19] SAP correlated with a higher total all-cause cost.

It was no surprise that SAP cases necessitated a higher material allocation. As there is no clear way to prevent an AP case from following a severe disease course and lowering the financial burden, we recommend conducting large population epidemiological studies to clearly stratify the citizens who are at risk of developing AP, or, more precisely, SAP. This can be followed by closely monitoring these cases through primary care medicine and/or cognitive behavioural therapy, as some of the causes are linked to modifiable risks, in order to help reduce the risk of developing this disease and bear the risk of necessitating hospitalisation.

As for the morphology analysis, we found the lowest median DCH in pseudocyst cases (USD 187.0), followed closely by interstitial (USD 190.4), and the highest DCH was in ANC (USD 225.2). There was a statistically marked discrepancy between interstitial and the following morphologies (in descending order): ANC, APFC, and normal pancreas. A statistically meaningful variation was also found between ANC and pseudocysts. This was not unexpected, as local complications, particularly necrosis [20], are correlated with mortality and severity, longer hospitalisation, ICU stay, and interventions, and, hence, higher costs [21].

One limitation of our study resides in the fact that we did not account for infected necrosis, a complication that can raise the cost of the case, not only due to specific pharmacological, surgical, and/or endoscopic intervention, but also due to being an important stimulus for starting systemic inflammatory response syndrome. Although there is no clear method for preventing the development of necrosis, procalcitonin screening and fine needle aspiration culture with an antibiogram might hinder the extension of infection and should, at least in theory, be more feasible for simple pathogen-oriented antibiotic therapy. Several cost-competitive antibiotics such as carbapenems might successfully be used in conservatively managing infected necrosis in selected patients [22].

With respect to the aetiology, we found the lowest median DCH in alcoholic-related AP (USD 169.2) and the highest in biliary-related AP (USD 235.4). We observed statistically substantial disparities between alcoholic-related and all other aetiologies groups in the study, with respect to a low significant difference between alcohol and the diabetes mellitus group. In addition, statistically significant deviations were found between biliary and all three other groups. Although there were no significant differences between the hypertriglyceridemia and diabetes mellitus groups, biliary aetiology seemed to be an independent risk factor for a higher cost of AP episodes. A 2016 retrospective Swedish study [23] conducted on a similar population also found a higher cost of biliary AP in comparison with a general cost of AP [5]. The authors implied that the same admission cholecystectomy might help cut costs for biliary-related AP. We will seek to investigate this hypothesis in our population in a future study, as we find it important for cost-reduction policies.

A statistically meaningful correlation was found between age and median DHC, possibly related to frailty [24,25,26]. Although not surprising, there should be better monitoring of citizens of older age with regard to the main risk factors that might cause AP. Nationwide screening for alcoholism (together with public-funded cognitive behaviour therapy or any kind of addiction therapy), serum triglycerides, or HbA1C, should be easy to implement within primary or secondary medical care networks and might be able to lower the risk for almost half the cases, as we recorded the prevalence of the three causes in our population. Additional pilot studies should be conducted to assess whether this is the right policy for cost efficiency. Notably, we could not find any notable correlation between LoS and median DCH.

This study has several limitations, one of which is its retrospective cohort character, as it is based on EHRs and is unicentric. Although improbable—due to the fact that in our country there is an oligopoly of the public-owned insurer that provides fixed rates to all public-owned hospitals and to most private ones—there is a possibility of inter-hospital or inter-regional cost-related discrepancies. As with all other retrospective studies, we also had limitations regarding the adjustment of potential confounders, inability to control exposure, and incomplete available data (with the exception of the following variables: sex, age, LoS, ward of care, ICU admittance, and severity). These limitations should be addressed by large multicentric randomised clinical trials. Further, we call for meta-analysis on the same topic, at least with regard to the European population, as there is a scarcity of data regarding the cost of this frequently encountered gastrointestinal disease, as reported by Peery et al. [4].

## 5. Conclusions

We found the median DCH of AP in southern Romania to be USD 203.8 and the median TCH to be USD 1360.5. An estimation of the total yearly cost of AP in Romania at around USD 19 million was obtained based on the statistical data. To our best knowledge, this is the first study of the cost of AP in Romania and probably one of the few originating from Eastern Europe. Drivers of increased costs of AP are older age, ICU, intra-hospital mortality, SAP, local complications such as ANC, biliary aetiology, and female sex. There is large heterogeneity and scarcity regarding cost-related data in the literature. We advocate for standardisation in reporting these data and for a call to action to all pancreas-focused researchers to improve the data available regarding costs worldwide. There are several policies that we believe might be cost-efficient in the setting of Romania, such as nationwide screening for risk factors (alcoholism, serum triglycerides, HbA1C) and addressing them through primary or secondary healthcare networks, especially in older age populations. We also recommend better stratification of the necessity of ICU on a case-by-case basis, as this is a large cost driver. Further, procalcitonin monitoring with fine needle aspiration cultures with antibiograms might help if they are deployed as soon as possible. These suggested policies should be implemented initially as pilot studies to assess their real cost efficiency.

## Figures and Tables

**Figure 1 healthcare-11-02482-f001:**
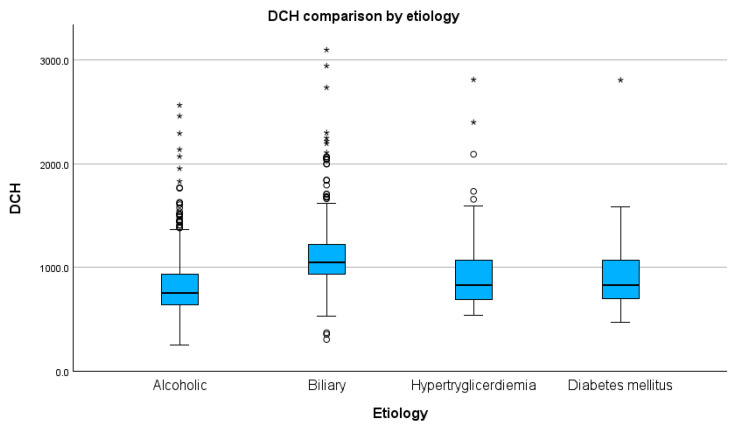
Comparison of median DCH by aetiology (values in RON). Circles indicate mild outliners (1.5–3.0 times IQR below Q1 or above Q3) while asterisks indicate extreme outliners (>3.0 times higher IQR below Q1 or above Q3).

**Figure 2 healthcare-11-02482-f002:**
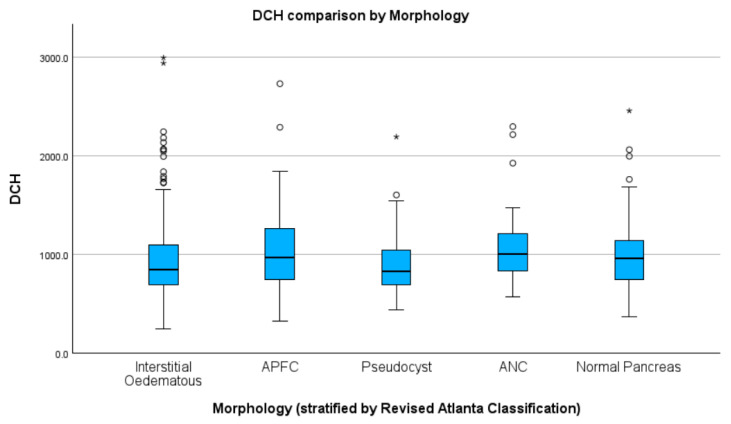
Median DCH compared by morphology (values in RON). Circles indicate mild outliners (1.5–3.0 times IQR below Q1 or above Q3) while asterisks indicate extreme outliners (>3.0 times higher IQR below Q1 or above Q3).

**Figure 3 healthcare-11-02482-f003:**
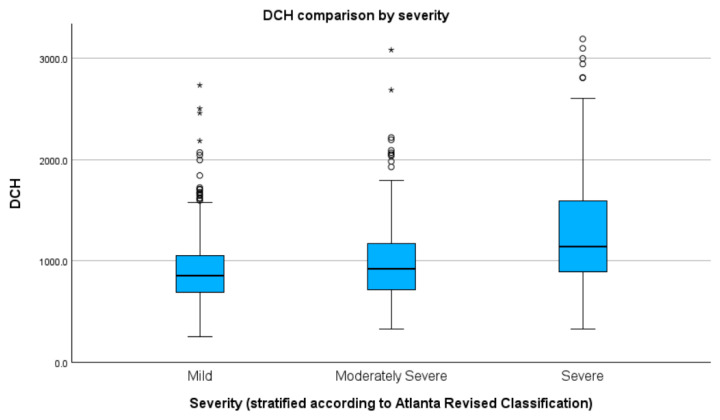
Median DCH stratified by severity (values in RON). Circles indicate mild outliners (1.5–3.0 times IQR below Q1 or above Q3) while asterisks indicate extreme outliners (>3.0 times higher IQR below Q1 or above Q3).

**Figure 4 healthcare-11-02482-f004:**
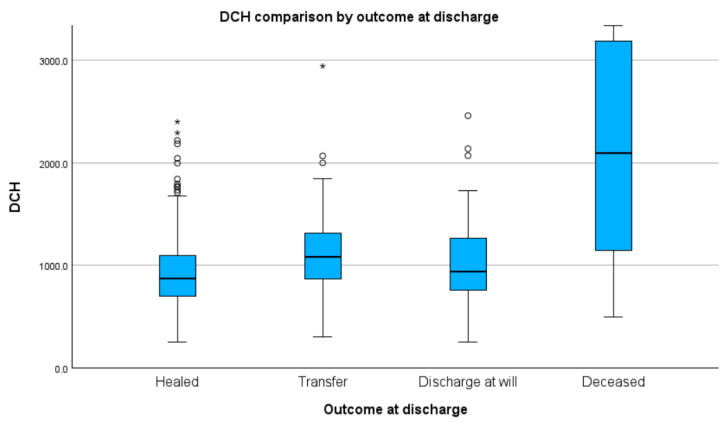
Median DCH by outcome at discharge (values in RON). Circles indicate mild outliners (1.5–3.0 times IQR below Q1 or above Q3) while asterisks indicate extreme outliners (>3.0 times higher IQR below Q1 or above Q3).

**Figure 5 healthcare-11-02482-f005:**
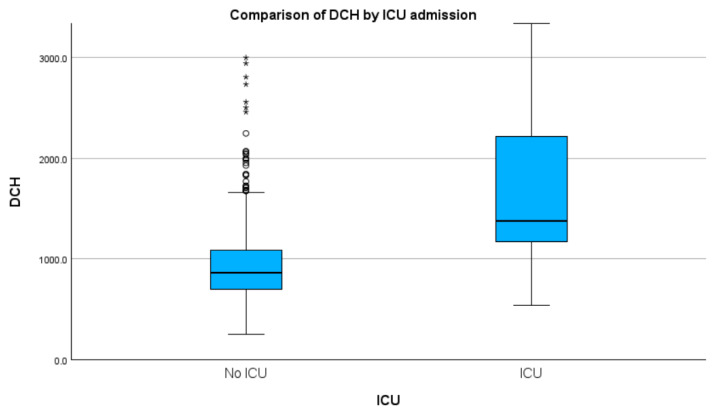
Median DCH grouped by ICU admission (values in RON). Circles indicate mild outliners (1.5–3.0 times IQR below Q1 or above Q3) while asterisks indicate extreme outliners (>3.0 times higher IQR below Q1 or above Q3).

**Figure 6 healthcare-11-02482-f006:**
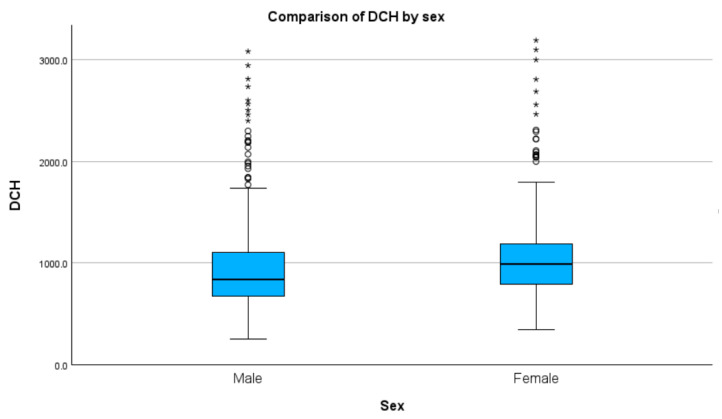
Median DCH by gender (values in RON). Circles indicate mild outliners (1.5–3.0 times IQR below Q1 or above Q3) while asterisks indicate extreme outliners (>3.0 times higher IQR below Q1 or above Q3).

**Figure 7 healthcare-11-02482-f007:**
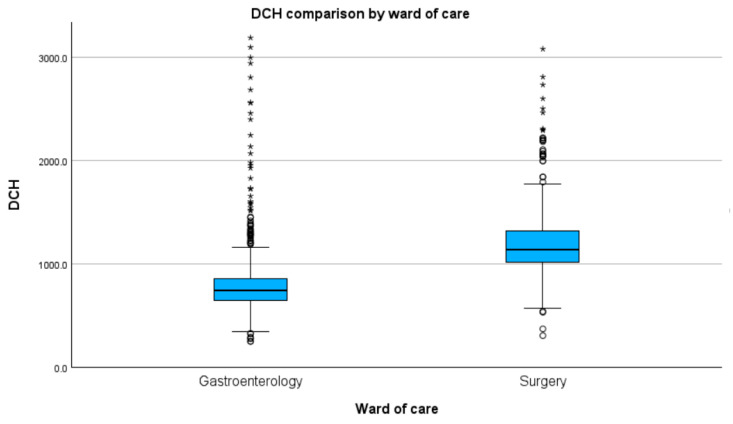
Median DCH by ward of care (values in RON). Circles indicate mild outliners (1.5–3.0 times IQR below Q1 or above Q3) while asterisks indicate extreme outliners (>3.0 times higher IQR below Q1 or above Q3).

**Table 1 healthcare-11-02482-t001:** General characteristics of the cohort.

Characteristics		Cases	Percent (%)	Median TCH (USD)	Median DCH (USD)
Department	Gastroenterology	860	58.4	1153.3	167.3
	Abdominal surgery	613	41.6	1804.3	256.5
Sex	Male	911	61.8	1245.0	188.7
	Female	562	38.2	1529.0	223.0
ICU	ICU-N	1348	91.5	1278.2	194.3
	ICU-Y	125	8.5	3937.9	309.0
Outcome	Healed/ameliorated	1234	83.8	1425.3	195.8
	Stationary	7	0.5	613.8	196.2
	Transfer	54	3.7	1151.4	243.7
	Discharge at will	101	6.9	700.6	211.1
	Aggravated	0	0	N/A	N/A
	Deceased	77	5.2	1310.3	469.4
Severity	Mild	758	51.8	1163.0	191.2
	Moderately severe	542	36.8	1509.5	207.4
	Severe	173	11.7	2220.5	256.7
Morphology	Interstitial	575	39.0	1360.6	190.4
	APFC	114	7.7	1909.0	217.3
	Pseudocyst	60	4.1	1328.6	187.0
	ANC	63	4.3	2132.4	225.2
	WON	4	0.3	1361.6	161.7
	Normal pancreas	210	14.3	1228.0	216.3
	N/A	447	30.3	1205.0	213.4
Aetiology	Alcohol	517	35.1	1147.1	169.2
	Biliary	509	34.6	1650.7	235.4
	Hypertriglyceridemia	80	5.4	1624.2	187.0
	Diabetes mellitus	62	4.2	1198.4	185.6
	Other	84	5.7	1544.9	203.9
	Idiopathic	221	15.0	1154.8	194.0
Median TCH—general		USD 1360.5			
Median DCH—general		USD 203.8			

Abbreviations: ICU—intensive care unit; APFC—acute peripancreatic fluid collection; ANC—acute necrotic collection; WON—walled-off necrosis; N/A—non-available; TCH—total cost of hospitalisation; DCH—daily cost of hospitalisation.

## Data Availability

Data available upon request to the corresponding author.

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
