# Peer review of "Acute Pancreatitis—Drivers of Hospitalisation Cost—A Seven-Year Retrospective Study from a Large Tertiary Center"

_healthcare, 2023, doi:10.3390/healthcare11182482_

Round 1

Reviewer 1 Report

The authors in their paper present a study of the burden of acute pancreatitis (AP) on healthcare services globally and outline the lack of cost-related studies in Romania.

(1) The introduction efficiently sets up the rational for the study, but there are several aspects that could be enhanced for better clarity and impact. The section mentions a lack of previous studies regarding the cost of AP episodes in Romania, but it would be beneficial to emphasize more strongly why this gap is significant.
(2) How does this gap hinder healthcare decision-making or patient outcomes?
(3) Maintaining a consistent tone throughout the introduction, balancing technical language with accessibility. Avoid overly complex sentence structures that may impede clarity

The "materials and methods" section provides a comprehensive overview of the study design and methodology. Overall, the section is well-structured and informative, but there are some aspects that could benefit from clarification and refinement. (1) Please check for consistency in terminology and formatting throughout the section. (2) Ensure that abbreviations are defined upon first use (This is the comment for the abstract). (3) Review the text for grammatical errors and sentence structure to improve readability. (4) Consider restructuring some of the longer sentences for enhanced readability and clarity. Cohesive sub-section heading could further improve the flow and organization of the content.

Results:
The results section provide the findings of the study. Overall, this section of the manuscript is well written.

Discussion:
This section provides a comprehensive analysis of the study's findings and comparisons with existing literature. While the section contains valuable insights, there are opportunities to refine the presentation and enhance the clarity of your arguments. (1) When referencing studies from the literature, briefly explain how these studies relate to your findings. (2) Why are you comparing your findings with these particular studies? (3) What insights do these comparisons provide? (4) How can you connect your findings to potential implications for healthcare practices, policies, or future research? (5) Avoid repetition.

The manuscript is well written, the quality of the English is well acceptable, however some long sentences should be restructured and rephrased.

Reviewer 2 Report

Thank you for the opportunity to review the present work by Pahomeanu et al. This subject is relevant to the readers and useful for future work. I have some comments/suggestions regarding this paper.

1. Have the English reviewed by  a native speaker, this needs editing.

2. The lay out of the figures, size of the figures, legends, numbers on the X and Y axis need extensive editing and is currently insufficient.

3. An import issue that needs further clarification is the study population, and whether this population impacts the generalizability of the findings:

- Please comment on the population in this hospital. Is it a tertiary referral center? Were patients primarily admitted from the ER, or are transferred cases also included?

- how many patients had interstitial pancreatitis and how many necrotizing pancreatitis? How many patients developed infected necrosis? How many interventions were performed, either transgastric, surgical or radiological?

4. The abstract contains too much details which limits readability

5. The introduction reads more like a summary of previous work by others.

6. Figure 1 is not relevant, this is not an actual flow-chart

7. A crucial question is: how were the cost per patient calculated? What methodology was used?

8. I am not aware of disregulation of diabetes being an important cause of pancreatitis. Please re-classify, and use the IAP/APA guideline for diagnostic criteria of biliary and alcoholic origin.

9. Was same admission cholecystectomy performed in biliary cases? Are these costs included in the present study?

10. What contributed most to the hospital costs? 

11. Did all patients undergo CT scanning?

12. Please to not classify a result with a p-value of > 0.05 as 'weakly significant'. 

13. Please classify pancreatitis severity according to the revised Atlanta criteria. According to table 1 11.7% was classified as severe (i.e. persistent organ failure) but only 8.55 was admitted to ICU. Please clarify.

14. What were the criteria to admit a patient to the gastroenterology versus surgery ward?

please see above

Round 2

Reviewer 2 Report

the authors have proved a swift and adequate reponst te reviewer questions, and the paper has improved markedly.